# Walking Forward Together—The Next Step: Indigenous Youth Mental Health and the Climate Crisis

Michael Brown, Sabina Mirza * , Jay Lu and Suzanne L. Stewart

Waakebiness Institute for Indigenous Health, Dalla Lana School of Public Health, 155 College Street Suite 400, Toronto, ON M5T 3M7, Canada; stormbringer.brown@utoronto.ca (M.B.); xinyi.lu@mail.utoronto.ca (J.L.); suzanne.stewart@utoronto.ca (S.L.S.)
* Correspondence: sabina.mirza@utoronto.ca

**Abstract:** The climate crisis has resulted in mental health challenges for varying demographic groups of all ages, but Indigenous youth are one of the most vulnerable populations impacted by the climate crisis. Conversations regarding Indigenous youth and the climate crisis are often held without their presence or input, identifying a gap in research and the literature. The findings from this review include the components of climate change regarding the mental health of Indigenous youth as being direct and indirect pathways of impact and resistance. Direct pathways include the more immediate and physical consequences of climate change associated with mental unwellness. Indirect pathways include less obvious consequences to those without lived experience, such as disruptions to culture and magnified social inequities, which also result in negative mental health consequences. The resistance component explores how Indigenous youth have been protesting and actively speaking out, which highlights the importance of the inclusion of Indigenous youth voices in decision-making spaces related to mental health service resources (i.e., funding) and policy in climate action. This review ends with a discussion on ways forward, the limitations herein, and how the uniqueness of the research may contribute to climate justice.

**Keywords:** indigenous youth; mental health; climate justice; resistance



## 1. Introduction

The climate crisis is a well-documented and urgent planetary health phenomenon and has been described as one of the greatest threats to human health in modern times, with significant effects on the mental health of young people as it is their future that is at risk [1–7]. Protecting adolescent and youth mental health while promoting positive youth development is important for the survival of all human beings, especially in the face of various societal issues such as the climate crisis [8]. Research studies show that young people most affected by climate change grief and anxiety are those who have directly experienced climate-related catastrophes and degradation, which continues to impact their current land and lifestyle, divagating from holistic health and well-being [5]. Those living in the Global South or belonging to Indigenous groups are often the ones feeling significant psychological distress around climate-related issues [2,5]. However, grief and anxiety are only the tip of the ever-melting iceberg that is the impact of climate change on the mental health of Indigenous youth. Indigenous youth are gravely impacted as they are the next generation of leaders and decision-makers and are already facing the consequences of environmental change and will continue to face its ongoing negative impacts. However, Indigenous youth are dismissed and often barred from decision-making spaces where real change can be brought forth, both in climate change policy and mental health services, despite being directly and indirectly impacted.

For context, the research question which this literature review aims to answer is: What are the mental health impacts of climate change for Indigenous youth, and what services are needed to support youth? This question was developed in a pilot project situated

in Tkaronto, Ontario, Canada (Treaty 13), and the Northwest Territories (NWT), Canada (Treaty 8), in which Elders and youth engaged in Indigenous Talking Circles [9] to develop themes around the mental health impacts of the climate crisis on Indigenous youth; these themes were then developed into creative drama plays by the youth that were shared at a national Indigenous mental health symposium in Tkaronto in the Fall 2022. Also included in the pilot was a survey in partnership with 2-Spirited People of the First Nations to establish baseline data on Indigenous youth mental health status and its intersections with the climate crisis. The results of both data sets from the pilot project directly inform this research question and the overall research approach, with partnership activities established at the pre-proposal stage.

This literature review will seek to answer these questions through an exploration of research about the impacts of the climate crisis on the mental health and well-being of Indigenous youth, with a goal of better understanding how youth can be empowered and supported in this context. Existing literature on climate change and health is narrowly focused on individual physical consequences, despite many known impacts to holistic individual and community well-being [10]. Developmentally, adolescence is a recognized time of significant prognostic implications, with long-lasting impacts on holistic health, on an individual and community level [11]. A delay and neglect in addressing these disruptions to development and the health needs of Indigenous youth often leads to severe and fatal consequences, such as suicide, crime, and general increases in mortality [11]. Therefore, the literature being reviewed will explore the impacts of climate crisis on the mental health and well-being of Indigenous youth from a culturally informed and holistic perspective.

For this literature review, the authors operationalize a definition of the climate crisis stemming from existing and current research and scholarship and through Indigenous Ways of Knowing. According to current research, the climate crisis can be related to human activities, comprised mostly of the extraction and consumption of fossil fuels, harming the Earth's biosphere. Such activities can result in rapid and increasing temperature changes, increasing and unprecedented rates of species loss, and ecosystem destruction [12,13]. However, in this context, the definition extends to include the imbalance of the Mother Earth/Human Beings relationship, which encompasses the prior definition plus the holistic nuances of human lifestyles and current bio-socio-political ontologies, epistemologies, methodologies and axiologies that are threatening not only our Mother Earth but also ourselves and our communities.

## 2. Materials and Methods

This literature review is integrative, critical, and includes a thematic analysis of the chosen articles. Some of the articles the authors have used throughout are not included in the results and analysis section, but are used to contextualize the work in terms of definitions and explanation of the climate crisis more generally.

This analysis began with a survey of the academic landscape to determine the organization of the conversations around Indigenous youth mental health and the climate crisis, abstracted to the point of looking at the organization of discourse regarding the impacts of climate change, adapting said discourse for this context. From there, the literature was included if it contributed to the areas of interest from the research questions (those being the intersections of climate change impacts, Indigenous youth, and mental health). To find the most accessible articles for this review, Google Scholar was used. The following search terms were used: "climate crisis and mental health", "climate crisis and Indigenous youth", "solastalgia and Indigenous youth", "solastalgia Canada", "climate crisis impact Indigenous mental health". References were collected and reviewed to identify whether the articles were eligible for closer review. The articles were further narrowed down based on the relevance to the impacts of climate change on Indigenous youth. The literature was included based on the relevance to Indigenous populations of Turtle Island (Canada) and that had mention of Indigenous youth, although international perspectives were included to provide a wider scope, allowing space for future inquiry. Articles were also included if

published between 2015 and 2023 (only for main articles), with a preference for scoping or literature reviews. The rationale for this range of dates is to work with the most recent publications while leaving enough of a longitudinal element to capture the ongoing scholastic conversations that occur. Articles were excluded if there was no discussion or mention of impacts of climate change on mental health and if the article was not freely available to the public. The article search was concluded once there was sufficient cohesion in the narrative and the content sufficiently overlapped in answering the questions. Supporting articles in the analysis were determined through the references made in the main articles. Overall, 9 peer-reviewed main articles and 20 peer-reviewed supporting articles were included in the analysis with 9 additional supporting articles (not included in the analysis) to provide context, examples, or foundational information. The article search is visualized in Figure 1.

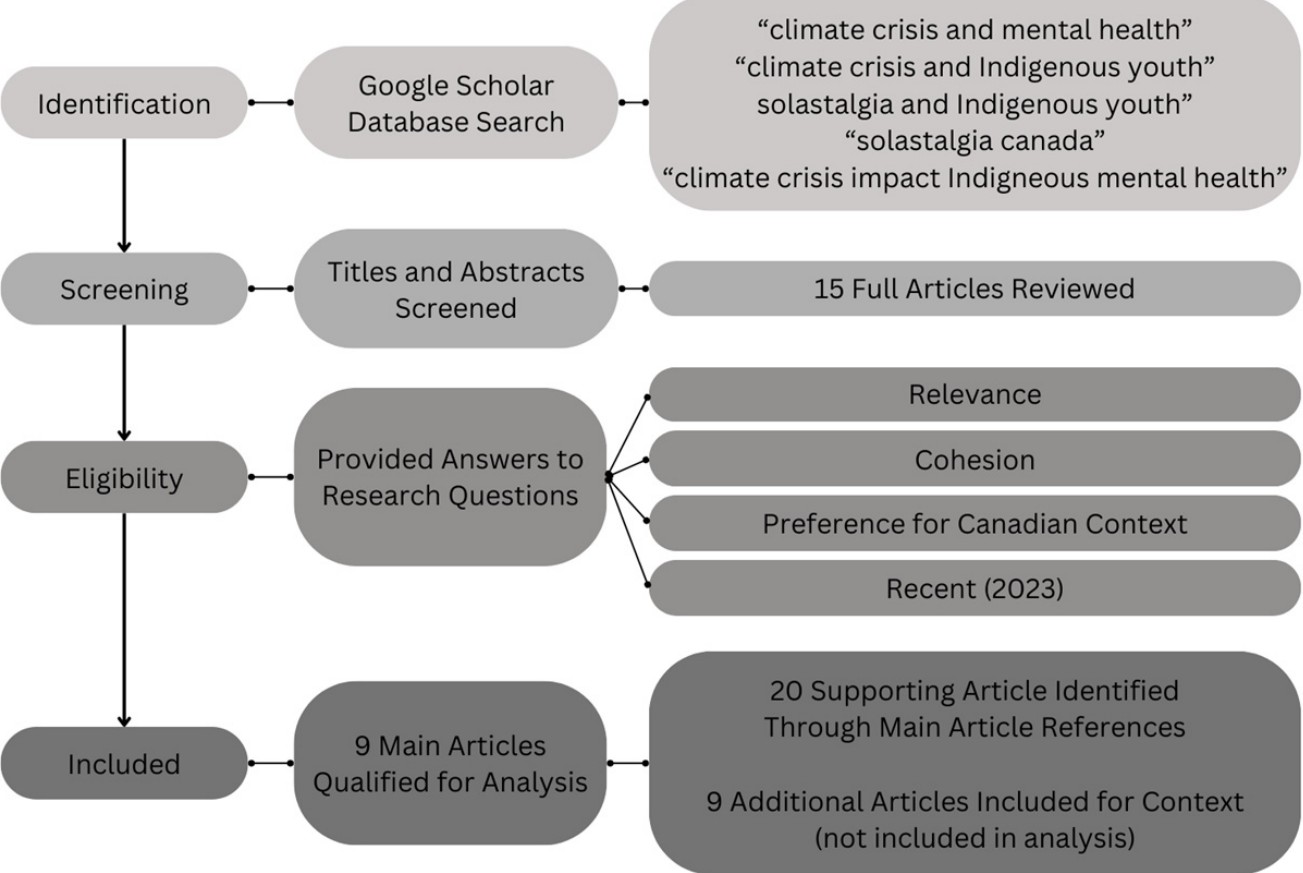

**Figure 1.** Diagram of article search methods for inclusion in analysis.

All eligible articles for this analysis were reviewed for the purpose of determining the structure by which the impacts of climate change on mental health have been written about—viz. determining the meta-themes. Once the meta-themes were identified, the literature was reviewed again for the purpose of organizing the content of each article into the respective meta-theme, thereby establishing themes, and examples within each meta-theme. Once the results of the analysis were determined, questions arose from team discussions around the context needed to tell the story, where individual articles were gathered for this purpose.

### 3. The Issues

Three main themes emerged from the literature review: direct pathways of impact, indirect pathways of impact, and resistance. Overall, the direct pathways of impact included changing temperatures and worsening weather conditions. The indirect pathways of impact included disruptions to the land and disruptions to infrastructure. Within disruptions to the land were the subthemes of cultural disruption and food insecurity. Within disruptions to infrastructure were multiple exasperated social inequities and amplified effects from colonization. Discussions related to resistance encompassed how Indigenous youth have been actively speaking out and seeking ways to adapt to climate change. As these impacts from climate change are being interpreted through a holistic lens, many of the factors overlap and weave together.

### 4. Direct Pathways of Impact

Direct pathways are more immediate physical impacts of climate change and include consequences of weather changes and shifts in ecological patterns that have remained relatively consistent over time. Multiple articles reported that rising temperatures [10,14] and severe weather events such as cyclones and floods [6,10,15,16] lead to increased frequency of injury, death, and trauma for individuals and communities. For example, in the context of the Northwest Territories in Canada, Lebel et al. [17] reported changes in rivers and lakes such as currents, water levels, and ice dynamics, and changes in wildlife such as vegetation and fish quality all directly result from climate change.

Another example of direct pathways of impact found in the literature was from Galway et al. [1] who described how significant warming in Canada has affected seasonality, permafrost, wildlife, sea ice, and extreme weather patterns. In the context of Indigenous youth mental health, the impacts of climate change, including the more severe weather and societal events, were reported to amplify any serious mental health challenges in addition to feelings of loss, displacement, distress, and hopelessness [6,17,18] In other words, direct pathways of impact elicit negative physical, emotional, and mental responses in Indigenous youth. Direct impacts on the spiritual elements of Indigenous youth well-being were not effectively addressed.

Galway et al. [1] reported that changes to the land and environment disrupted Indigenous Peoples, including youths, from accessing culturally significant geography and teachings, interrupting daily living, subsistence, land relations, culture, and health. The unpredictability of the land and other barriers to travel were also reported to endanger the role of land-based activities in fostering and maintaining social relationships [17]. For example, Lebel et al. [17] reported that a significant number of Inuit and Inupiat communities felt trapped, stressed, and anxious from being confined to their communities for longer periods of time due to unpredictable travel routes. Furthermore, trauma from injury and extreme weather events were found by Cunsolo Willox et al. [10] to worsen negative feelings, such as concern for the safety of individuals and their families.

Additionally, the inability to access the land for historical, spiritual, healing, hunting, fishing, or other holistically significant activities were noted as potential fuel for a swathe of emotional responses, including anxiety, fear, stress, grief, and a sense of loss and anger [10,19,20]. For example, Vecchio et al. [6] found that youths in Rigolet, Canada, were fearful for the futures of their cultural identities resulting from the diminishing connection to land. Climate change was reported to amplify current and historical traumas, including land dispossession, forced relocation, intergenerational trauma, and loss of cultural knowledge exchange [10]. The need to relocate and the burden of decreased mobility was found to be associated with feelings of loss and fear [6].

It was found within the literature that mental health was also impacted by the climate crisis through the lack of agency and self-determination of the communities and individuals that are more vulnerable (i.e., Indigenous communities), as the lack of agency may lead to feelings of depression or hopelessness [10].

### 5. Indirect Pathways of Impact

Indirect pathways are more holistic and are influenced by the interconnected social systems that exist for Indigenous youth beyond the biomedical model [1]. The indirect pathways of impact are found at the constantly fluctuating intersections of culture, social equity, infrastructure, and mental health. Indirect pathways of impact were also described in the literature in the form of disruptions to the land and environment and disruptions to infrastructure, each with overlapping impacts on mental health and wellness.

The literature described the impacts of climate change on Indigenous youth and the connections between land, community, and health as being present culturally, geographically, and socially [21–23]. Witnessing the changes in land and climate were reported to contribute to the mental health impacts on Indigenous youth, as trauma and other related health challenges often ensued [22,24–26].

Disruptions to food security were mentioned across the literature as indirect pathways that negatively impact mental health, as disruptions to hunting routes, fishing areas, and changes to biodiversity and the land continue to result from climate change [6,10,22]. Moreover, Cunsolo Willox et al. [10], found that anxiety and depression may also stem from the more frequent need and reliance on store-bought foods that are often more expensive and less nutritious than traditional food sources.

Lebel et al. [17] described the impacts of climate change on the access of traditional country foods in multiple Indigenous contexts whereby the family and community time spent sharing traditional foods was undermined by the lack of access and disruption to wildlife on the land and in the sea. The outcomes of those impacts included feelings of discouragement, loss, disconnection to culture, decreased confidence, and decreased community strength. Vecchio et al. [6] described international perspectives of the impacts of climate change on food security as having added barriers to health and well-being. For example, social elements of traditional harvesting were less available for managing stress; insecure access to traditional foods magnified financial distress of families; and increased safety concerns for community members while travelling and finding new routes.

Cunsolo Willox et al. [10] reported that infrastructural disruptions, such as melting ice roads and rising sea levels, further impacted mental health and well-being in addition to damages to pipelines, water-treatment facilities, and bridges. The authors found that these disruptions added to the mental stress of existing drinking water challenges, housing and homelessness challenges, and health care access, which elevated anxiety and distress from anticipatory or current loss and/or displacement.

Across multiple publications, authors agreed that environmental changes, including resource extraction and environmental pollution, mirrored and often magnified the existing effects of colonization [10,17,22]. The fear of further losing connection to culture and culturally significant means of fostering individual and community well-being were salient throughout the literature [6,10,17,18].

Vecchio et al. [6] reported that disruptions in infrastructure mirrored the systemic colonial mechanisms that reinforced barriers for Indigenous Australian employment, negatively influencing the capacity to provide, which leads to negative mental health outcomes. Furthermore, it was described by the authors that the lack of employment opportunities reinforced a pressure to assimilate to the colonial systems—the same systems that continue to silence their voices (see, for example, https://www.theguardian.com/australia-news/2023/oct/14/australia-rejects-proposal-to-recognise-aboriginal-people-in-constitution, accessed on 30 October 2023) [27].

### 6. Resistance

Current literature suggests that the importance of supporting the mental health of Indigenous youth has grown in prevalence since the beginning of the COVID-19 pandemic as youth have been facing additional unpredictable lifestyle changes [22]. The literature reported that Indigenous youth are experiencing what scientists have deemed 'Solastalgia',

or ecological grief [22]. Solastalgia has been connected to feelings of anger, frustration, and helplessness [22,28].

Youth were reported to quickly organize and powerfully engage in societal decision-making, especially when it comes to their futures [22,29]. For example, Lebel et al. [17] reported that Indigenous youth have been speaking up in spaces where they have been historically uninvited, such as United Nations Climate Change conferences. The authors described how it not only demonstrates their awareness, but also their desire to be involved and to engage in self-advocacy. The authors also described other adaptive responses from Indigenous youth including learning traditional crafts and skills, participating in sports, working, spending quality time with family and friends, participating in music activities, going to youth centers, and taking walks around town.

Youth were holding school strikes, highlighting concerns about the climate, and making a call to be heard and taken seriously by authorities around the world [22,30]. It was also reported that youth are advocating for themselves to be included in conversations of ecological degradation and human-induced destruction and extraction [4,22]. Noronha et al. [31] discussed resistance factors for Indigenous youth regarding the climate crisis, including culture, language, personal agency, and participating in traditional ceremony. Gislason et al. [22] found that when the voices of Canadian youth were included in community climate initiatives, they gained a sense of control, governance, representation, and autonomy. The literature found that youth tend to become more empowered through self-determination actions, such as local activism, self-expression, and inclusion in research and policy [22,28]. Authors reported that when Indigenous youth have opportunities to weigh in on decisions involving resources, being members of participatory research processes, and having social networks, they feel empowered and resilient [31].

Lebel et al. [17] described the increases in confidence and connection to land for youth involved in political advocacy and climate awareness activities. Noronha et al. [31] found that when communities have the authority to self-determine solutions, methods of healing, and ways to promote wellness, the self-efficacy of individual community members is simultaneously increased. Additionally, Vecchio et al. [6] reported that youth in Rigolet, Canada, have found mental resilience, preparedness, and adaptability in the recognition of the community-level capacity to adapt and survive [6,14]. Furthermore, including and listening to Indigenous youth in spaces where decisions are made was reported to empower and support the self-determination of Indigenous youth [1,31].

## 7. Discussion

Findings from the literature review about the mental health impacts on Indigenous youth were centralized around direct pathways of impact, indirect pathways of impact, and resistance. The direct pathways of impact included more immediate and physical consequences of climate change such as floods and other extreme weather events, which compromised physical safety and were a source of worry and added stress. The indirect pathways of impact included the consequences of climate change such as cultural disruption and food insecurity, leading to trauma and other negative mental and emotional reactions. The resistance of Indigenous youth focussed on the intention to be included and taken seriously in decision-making spaces such as government, politics, and the self-determination of Indigenous Peoples. The overall findings are summarized in Figure 2 (see below). In this Figure, you will find the visual descriptions of the direct and indirect pathways of climate change impacts on the mental health and well-being of Indigenous youth. The figure uses blocks to illustrate the difficulty of teasing out the complexities and nuances of the interplay of contributing factors. You will also notice the element of resilience depicted as a solid arc, supported by factors of resistance, with many factors continuing beyond resilience, indicating an urgent need for additional support. Finally, you will notice the embedded nature of Indigenous youth mental health and well-being within the individual and community health block, indicating the interconnectedness of individuals and community health.

The strengths of the literature included the focus on mental health impacts of climate change on Indigenous Peoples from a global perspective, highlighting the often-ignored consequences of climate change, especially on culture and well-being. This is a strength that connects international Indigenous communities rather than isolating them, where community and research partnerships can help nurture intercommunity relationships and strategies to rebalance the natural world. Moreover, the inclusion of Indigenous perspectives that may not adhere or conform to the mainstream biomedical model of health and wellness, including elements of physical, emotional, mental, and spiritual well-being, also creates foundational inquiry for the creation of decision-making spaces that intend to include the voices and authority of Indigenous people and youth.

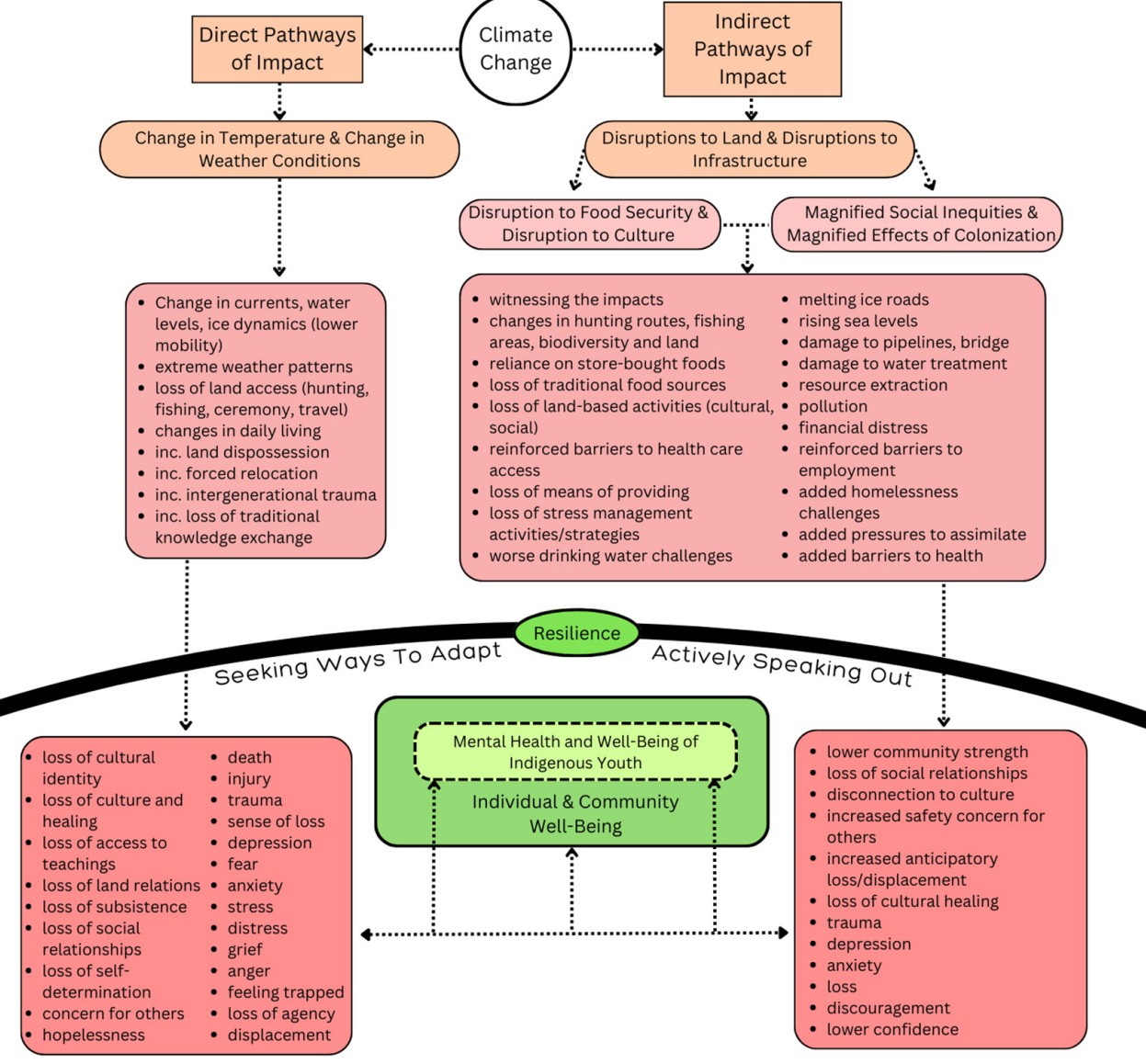

**Figure 2.** Direct and indirect pathways of climate change impacts on the mental health and well-being of Indigenous youth.

A gap in the research is that it is often conducted using a deficit-based lens, seeking the gaps and pitfalls of the current situation. Although it is useful to know what weaknesses exist and to better understand the context and impacts, deficit-based approaches tend to add stress through the narrative of hopelessness. A weakness of the literature was the tendency for research to focus more on perspectives from adults and on impacts that were not in

the control of Indigenous youth, such as damage to infrastructure. This is another clear indication that Indigenous youth need to be included, heard, and supported in the decisions that impact them. Another gap in the literature was the tendency to focus on Indigenous communities that were remote, as opposed to urban. This is a limitation because of the greater chances of cultural homogeneity in smaller, more remote Indigenous communities, when compared with the cultural diversity in urban settings, such as Tkaronto (Treaty 13). There needs to be further inquiry regarding the perspectives of Indigenous youth on food security, particularly in the context of learning about and having access to the traditional means of food security.

Additionally, it is essential to cultivate work that addresses local climate issues in the context of cities, rural, remote, northern, island, and traditional territory community contexts—conducting research and community-based climate actions not only for children and youth, but with them. It also calls for integrating findings into future policies and practices to build capacity in the mental healthcare sector in Canada by integrating mental health support into existing healthcare facilities and programs. This is particularly important in rural and remote communities with limited mental health programming and funding. As mentioned in the literature, mental health services are potentially another form of cultural assimilation, as health and wellness are not in the control of the communities being affected; they continue to be structured and designed through the western lens and understanding of health [10].

The issues discussed in this literature review indicate that there is a clear need for the respectful and meaningful involvement of Indigenous youth in an array of decision-making spaces. For example, the direct pathways of impact inspire inquiries about the perspectives of Indigenous youth on their experiences of interacting with primary health services, such as hospital visits and mental health crisis resources. The indirect pathways of impact indicate a need for the perspectives of Indigenous youth on the ways in which they maintain their connection to the land and barriers to doing so, from both remote and urban contexts. Furthermore, the indirect pathways indicate the need to include Indigenous youth in decisions regarding industry and policy (i.e., resource extraction, infrastructural development, city planning, etc.).

In discussing limitations of the research from the literature review articles, it is also important to discuss the limitations that arise when completing a literature review more specifically. For instance, due to the limitations of time and space, only a handful of articles could be included within this review, as it is near impossible to ensure that all literature on a particular topic is included. For example, the decision to only include open access articles was made to provide accessibility in the information being used, such that the higher value is placed on accessibility than privilege (a socially constructed trade-off). In addition, although the method and process for completing the review was systematic, the authors acknowledge that qualitative research is also subjective and nuanced; therefore, the articles included did pertain to the authors original research questions, resulting in related themes and meta-themes, all of which can be perceived as limitations to the research and study as the researchers must narrow their focus and make specific decisions about what to include.

Importantly, the climate crisis is not an issue that is to be pushed onto the next generation, but rather, is a mixture of conversation and action in which Indigenous youth must be included so that the *current* generation is better able to support youth needs. The indirect pathways that magnify the impacts of colonization indicate the need to re-establish and reinforce intergenerational healing. For the cultural losses and social inequities that are mediated by the climate crisis, a healing approach could be the inclusion of multiple generations in systems solutions. Support of intergenerational inclusion and quorums being required to include Indigenous Peoples from all living generations would provide a clear demonstration of Indigenous reconciliation.

## 8. Conclusions

Findings from this literature review indicate that Indigenous youth are among the most vulnerable populations on the planet in terms of human health and the climate crisis and are negatively impacted by decisions being made by government and health care systems. Not only does current literature claim that the impacts on youth are immediate and physical (i.e., increased frequency of death), but also indicates existence of indirect impacts, such as cultural disruption, food insecurity, and magnified consequences of colonization. Ultimately, these pathways of impact have consequences on physical, emotional, mental, and spiritual well-being for Indigenous youth.

This review of literature has informed a research project that involves the voices of young Indigenous Peoples in Toronto and the Northwest Territories about how the climate crisis has affected their mental health. The project is also interested in the programmatic and cultural and ceremonial supports and services that are required to support the individual needs of Indigenous youth, thus adding to the originality of the research and its impact. Further outcomes include an analysis of the results of the research involving young people, with their perspectives included, further adding to the uniqueness and innovative nature of the research supported by the current review of literature.

The immediate shift that this work calls for is the inclusion of Indigenous youth representation and input in the conversations and decisions being made across society, including but not limited to: all conversations and decisions about services being provided (or in development to be provided) regarding mental health, physical health, emotional health, cultural health, community health, spiritual health, decolonization efforts; biomedical understandings and applications of health; food security services; infrastructural conversations and decisions such as homelessness services, access to and use of Land, pipeline and other geopolitical and resource use/land development projects, civil engineering and other city planning projects, other decolonization efforts not captured in the previous mention. All of these have been implicated in this review, as the synthesis of the findings indicate the need for Indigenous youth inclusion of all these conversations, as they are key stakeholders. Further, the diversity of Indigenous cultures and voices cannot be captured with a single Indigenous youth voice.

Indigenous youth are speaking out (i.e., school strikes and political activism) to be included in the discussions and decisions being made by community and political leaders, where doing so would shift the praxis of research and health care development and delivery to empower and support Indigenous communities and youth. Without this praxis shift, Indigenous communities and youth will continue to be marginalized by the health, social and political systems that operate around and for them. Indigenous ways of knowing and doing involves *walking together as multiple generations*, which means that all people need to have a voice in climate crisis issues and solutions.

**Author Contributions:** M.B., S.M., J.L. and S.L.S. contributed equally to all sections of the paper. All authors have read and agreed to the published version of the manuscript.

**Funding:** This research received no external funding.

**Institutional Review Board Statement:** Not applicable.

**Informed Consent Statement:** Not applicable.

**Data Availability Statement:** No new data were created or analyzed in this study. Data sharing is not applicable to this article.

**Conflicts of Interest:** The authors declare no conflict of interest.

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
