# Peer review of "Walking Forward Together—The Next Step: Indigenous Youth Mental Health and the Climate Crisis"

_psych, doi:10.3390/psych6010013_

Round 1
Reviewer 1 Report
Comments and Suggestions for Authors
This is a very interesting and important topic for a review and the article has a lot of potential, but it also needs a lot of work to be in publishable format.
The article completely lacks a description of the search strategy and the way that the authors sorted and analyzed what they found. There is no guiding question or set of questions in the intro. The intro is already making conclusions, which is inappropriate and then the text directly launches into results with no discussion of how the authors got there. You need a methods and approach section.
A figure showing how the various pathways work would be very helpful.
The articles they include seem relevant and interesting and their analysis of themes and gaps may be accurate, but without any sense of the process, it is very hard to judge.
The intro needs to be an intro - without conclusions already drawn. And you need to lead with why adolescent/youth health is an important period fall ALL teens/youth and then go into why indigenous youth need particular attention.
Also you never define climate crisis.
This article is a solid start toward being a strong and important contribution to the literature and more than that a call to action, but without better structure and communication it doesn't meet the mark.
Here are some tips on writing lit reviews:
https://writingcenter.unc.edu/tips-and-tools/literature-reviews/
Also one small note - at one point they say that a situation is 'exasperated' - the authors mean 'exacerbated'. Very different meanings for those 2 words.
Comments on the Quality of English LanguageThe English is overall pretty good, but it is too casual in spots. It seems to be written by multiple people. More formal academic writing in places and in other places more casual.
Author Response
Response to Reviewer 1
Journal: Psych (ISSN 2624-8611)
Manuscript ID: psych-2769260
Type: Review
Title: Walking Forward Together – The Next Step: Indigenous Youth Mental Health and the Climate Crisis
Section: Neuropsychology, Mental Health and Brain Disorders
Dear members of the review and editorial team,
The authors would like to thank you for taking the time to read through our paper and offering us suggestions on how to improve the paper so that it is more comprehensive. Below, you can find a point-by-point response to the comments you had provided:
Thank you
Sincerely,
The authors
Comments and Suggestions for Authors:
This is a very interesting and important topic for a review and the article has a lot of potential, but it also needs a lot of work to be in publishable format.
Thank you, we also believe it is a very important topic and therefore want to ensure it is published for knowledge dissemination purposes. We believe the edits included in the revised version will allow for the paper to be of publishable format.
The article completely lacks a description of the search strategy and the way that the authors sorted and analyzed what they found. There is no guiding question or set of questions in the intro. The intro is already making conclusions, which is inappropriate and then the text directly launches into results with no discussion of how the authors got there. You need a methods and approach section.
We have now included a methods section, which includes a description of the search strategy used for the review, in terms of how the articles were sorted and analyzed. As per your suggestions, we have now also included our research/guiding questions in the introduction. Please see our revised draft where we discuss our approach in the methods and materials section.
A figure showing how the various pathways work would be very helpful.
A figure showing how the various pathways work has now been included (see Figure 2)
The articles they include seem relevant and interesting and their analysis of themes and gaps may be accurate, but without any sense of the process, it is very hard to judge.
We have now incorporated further details about the process in hopes that the relevance of the articles and strength of the analysis of themes and gaps is contextualized.
The intro needs to be an intro - without conclusions already drawn. And you need to lead with why adolescent/youth health is an important period fall ALL teens/youth and then go into why indigenous youth need particular attention.
As per your suggestions, we have revised and edited our introduction to paint a broader picture of why adolescent and youth mental health is important in the context of social issues, such as the climate crisis, and then discuss why this issue is pertinent for Indigenous youth specifically. However, it should be noted that there are colonial implications made in the centering all teens/youth before Indigenous teens/youth, as the article is describing the need to center Indigenous youth voices. For these reasons, we have intentionally decided to focus on the impacts of the climate crisis on Indigenous youth mental health more specifically throughout the paper.
Also you never define climate crisis.
We have now operationalized a definition of climate crisis into our introduction section.
This article is a solid start toward being a strong and important contribution to the literature and more than that a call to action, but without better structure and communication it doesn't meet the mark. Here are some tips on writing lit reviews:
https://writingcenter.unc.edu/tips-and-tools/literature-reviews/
Thank you for your comments. We are excited to contribute this article to the literature. We have read, reviewed and incorporated a lot of what is shared in the link.
Also one small note - at one point they say that a situation is 'exasperated' - the authors mean 'exacerbated'. Very different meanings for those 2 words.
We have reviewed this small note, and regarding the word usage for the sentence reading “the situation is exasperated”, the authors intended to use the word “exasperated”.
Comments on the Quality of English Language
The English is overall pretty good, but it is too casual in spots. It seems to be written by multiple people. More formal academic writing in places and in other places more casual.
As there are multiple authors who have contributed to this paper, there may have been some discrepancies regarding the language usage. We have gone ahead and edited the paper as much as possible to ensure formal language is consistently used throughout.
Reviewer 2 Report
Comments and Suggestions for Authors
The following comments need addressing before considering it for publication:
1. Clearly declare your research questions.
2. The adopted methods are not adequately described in detail. They must be explained with nuanced details.
3. Specify the type of literature review conducted, along with the procedures, protocols, criteria, and datasets used.
4. Instead of using a wide variety of sections, it is suggested to follow the instructions and structure of a scientific paper.
5. Visualize your unique outcomes in the form of diagrams and flow charts.
6. Illustrate the frequency of obtained keywords in the form of a word cloud.
7. Graphically delineate the identified research gaps and implications for future studies.
8. If possible, provide a Sankey diagram for your evidence.
9. Discuss the limitations of your study.
10. Address the originality of your research in the conclusion.
Author Response
Response to Reviewer 2
Journal: Psych (ISSN 2624-8611)
Manuscript ID: psych-2769260
Type: Review
Title: Walking Forward Together – The Next Step: Indigenous Youth Mental Health and the Climate Crisis
Section: Neuropsychology, Mental Health and Brain Disorders
Dear members of the review and editorial team,
The authors would like to thank you for taking the time to read through our paper and offering us suggestions on how to improve the paper so that it is more comprehensive. Below, you can find a point-by-point response to the comments you had provided:
Thank you
Sincerely,
The authors
Comments and Suggestions for Authors
The following comments need addressing before considering it for publication:
- Clearly declare your research questions.
We have now included our research questions in the introduction. - The adopted methods are not adequately described in detail. They must be explained with nuanced details.
We have included a methods and materials section, where the search strategy and how articles were sorted and organized is explained in more detail than the original draft. - Specify the type of literature review conducted, along with the procedures, protocols, criteria, and datasets used.
The methods and materials section has now been updated to specify the type of literature review conducted and has hopefully now captured the procedures, protocols, criteria and datasets used. - Instead of using a wide variety of sections, it is suggested to follow the instructions and structure of a scientific paper.
We have revised the paper to use the sections and structure outlined by the journal guidelines. - Visualize your unique outcomes in the form of diagrams and flow charts.
Please see Figure 1 for a visualization of unique outcomes included in the diagram/flowchart. - Illustrate the frequency of obtained keywords in the form of a word cloud.
The authors have included a diagram illustrating the obtained keywords in Figure 1.
- Graphically delineate the identified research gaps and implications for future studies.
If this is pertinent to the revisions, would the reviewer be able to provide us with an example of what a graphically delineated visualization looks like with respect to research gaps and implications for future studies? Our study is mainly qualitative and therefore a sample would help us make this revision, if necessary for knowledge dissemination and publication. - If possible, provide a Sankey diagram for your evidence.
A version of a Sankey diagram has been incorporated into the methods section (see Figure 1) - Discuss the limitations of your study.
The authors have included a brief discussion of the limitations of the study, or doing a literature review on this topic, which is now embedded within the discussion section. - Address the originality of your research in the conclusion.
Our conclusion has been revised to address the originality of our research
Round 2
Reviewer 1 Report
Comments and Suggestions for Authors
Thank you for this very responsive review. The article is vastly improved and much easier to follow as a reader. With a few more tweaks it will be a very nice contribution to the literature and help to raise awareness of these extremely important issues.
I appreciated the additional detail you added regarding your search strategy and the figure that you included to describe the flow. In fact, I think the process you used really provides even more weight to the focus of your article and the conclusions you draw. I also appreciated the limitations section you added. That said, given that you limited your search to open access articles, you need to include at least a brief rationale for that decision as that likely means that you missed including some articles that would have been relevant to your topic. This detail doesn't appear in your figure - would be good to add and learn how many fell out of the search as a result. It would also be good to understand the rationale for the years you chose - just that they are most recent? I would add a phrase or sentence.
I would like to address the comment that the authors made in their response (below). I think they misunderstood what I was saying (and I take responsibility for not being more clear) - I was not suggesting centering this article on all teens/youth. I was trying to help the authors underscore their point further in the intro only - i.e. all teens/youth are at a vitally important stage where this age group has very particular health needs different from other stages of childhood and life that need focus and attention. And then within that extremely important stage, indigenous youth need even greater focus and care/support because they are grappling with (and can lead us all forward on) issues that are uniquely indigenous that require attention - e.g. impacts of climate change, including acknowledging harms and going forward with inclusion of their voices, resistance, and change. Hope that clarifies the point.
However, it should be noted that there are colonial implications made in the centering all teens/youth before Indigenous teens/youth, as the article is describing the need to center Indigenous youth voices. For these reasons, we have intentionally decided to focus on the impacts of the climate crisis on Indigenous youth mental health more specifically throughout the paper.
I like what you added to the conclusion but would suggest breaking up into multiple sentences the one that starts "This review of the literature has informed a research project..." It is very long and becomes difficult to follow, so the important points you are making get a little lost.
Author Response
Response to Reviewer 1 – Round 2
Journal: Psych (ISSN 2624-8611)
Manuscript ID: psych-2769260
Type: Review
Title: Walking Forward Together – The Next Step: Indigenous Youth Mental Health and the Climate Crisis
Section: Neuropsychology, Mental Health and Brain Disorders
Dear members of the review and editorial team,
The authors would like to thank you for taking the time to read through our paper once more and provide us with an opportunity for revisions to refine the paper even further.
Below, you can find a point-by-point response to the comments you had provided:
Thank you
Sincerely,
The authors
Comments and Suggestions for Authors
- Thank you for this very responsive review. The article is vastly improved and much easier to follow as a reader. With a few more tweaks it will be a very nice contribution to the literature and help to raise awareness of these extremely important issues.
We really appreciate your feedback and thank you for your encouragement. We have now attempted to make those few tweaks in hopes of contributing more comprehensively to the literature about the important issue of Indigenous youth mental health and climate crisis.
- I appreciated the additional detail you added regarding your search strategy and the figure that you included to describe the flow. In fact, I think the process you used really provides even more weight to the focus of your article and the conclusions you draw. I also appreciated the limitations section you added.
- That said, given that you limited your search to open access articles, you need to include at least a brief rationale for that decision as that likely means that you missed including some articles that would have been relevant to your topic. This detail doesn't appear in your figure - would be good to add and learn how many fell out of the search as a result.
Details have been added regarding the rationale behind including only open access articles. However, the specific amount that were excluded for this reason is unknown.
- It would also be good to understand the rationale for the years you chose - just that they are most recent? I would add a phrase or sentence.
Details have been added regarding the rationale for the date range of articles being included.
- I would like to address the comment that the authors made in their response (below). I think they misunderstood what I was saying (and I take responsibility for not being more clear) - I was not suggesting centering this article on all teens/youth. I was trying to help the authors underscore their point further in the intro only - i.e. all teens/youth are at a vitally important stage where this age group has very particular health needs different from other stages of childhood and life that need focus and attention. And then within that extremely important stage, indigenous youth need even greater focus and care/support because they are grappling with (and can lead us all forward on) issues that are uniquely indigenous that require attention - e.g. impacts of climate change, including acknowledging harms and going forward with inclusion of their voices, resistance, and change. Hope that clarifies the point.
- However, it should be noted that there are colonial implications made in the centering all teens/youth before Indigenous teens/youth, as the article is describing the need to center Indigenous youth voices. For these reasons, we have intentionally decided to focus on the impacts of the climate crisis on Indigenous youth mental health more specifically throughout the paper.
Thank you for clarifying and engaging in this discussion with us. With careful consideration, it seems that we are all on the same page about the points being made here and so we have kept the revisions made on the point of how Indigenous youths’ mental health is disproportionately affected by climate change, in comparison to other youth their age, as per your suggestions.
- I like what you added to the conclusion but would suggest breaking up into multiple sentences the one that starts "This review of the literature has informed a research project..." It is very long and becomes difficult to follow, so the important points you are making get a little lost.
The flow of the paragraph has been reorganized to parse out the sentences and points.
The clarity and flow should now be improved.
Reviewer 2 Report
Comments and Suggestions for Authors
The implemented revisions are mostly satisfactory. However, Sankey diagram still has not performed within the manuscript. Fig. 1 refers to a type od PRISMA flow chart. You may easily google it to figure out how to generate a Sankey diagram for your literature review.
Author Response
Response to Reviewer 2 – Round 2
Journal: Psych (ISSN 2624-8611)
Manuscript ID: psych-2769260
Type: Review
Title: Walking Forward Together – The Next Step: Indigenous Youth Mental Health and the Climate Crisis
Section: Neuropsychology, Mental Health and Brain Disorders
Dear members of the review and editorial team,
The authors would like to thank you for taking the time to read through our paper once more and provide us with an opportunity for revisions to refine the paper even further.
Below, you can find a point-by-point response to the comments you had provided:
Thank you
Sincerely,
The authors
Comments and Suggestions for Authors
- The implemented revisions are mostly satisfactory. However, Sankey diagram still has not performed within the manuscript. Fig. 1 refers to a type od PRISMA flow chart. You may easily google it to figure out how to generate a Sankey diagram for your literature review.
Thank you for your feedback. We reviewed Sankey diagrams on Google and the authors feel that a Sankey diagram would not be an appropriate form of data visualization due to the overlapping and confounding nature of the impacts and would not provide any actionable insights of the pathways between factors and their outcomes. However, as we believe the reviewers suggestions for adding diagrams to demonstrate the flow of the literature review is important, the authors have now included a diagram that we believe will provide insight and clarification of findings. Please review Figure 2 to see our newly implemented diagram.